# Interaction of Physical Activity and Personality in the Subjective Wellbeing of Older Adults in Hong Kong and the United Kingdom

**DOI:** 10.3390/bs8080071

**Published:** 2018-08-06

**Authors:** Bill Cheuk Long Chan, Michelle Luciano, Billy Lee

**Affiliations:** 1Department of Psychology, University of Edinburgh, Edinburgh EH8 9JZ, UK; michelle.luciano@ed.ac.uk (M.L.); b.lee@ed.ac.uk (B.L.); 2Centre for Cognitive Ageing and Cognitive Epidemiology, University of Edinburgh, Edinburgh EH8 9JZ, UK

**Keywords:** older adults, personality, physical activity, subjective wellbeing

## Abstract

Subjective wellbeing (SWB) has been widely accepted as one of the most important elements of successful ageing. The present study explores the impact of two well-established correlates of SWB: physical activity and personality. Physical activity and each of the Big Five personality traits are consistent predictors of SWB, but there has been little research on whether certain personality traits enhance or hinder the psychological benefits of physical activity in older adults. This study examines the interactions of leisure-time physical activity and personality traits on SWB, and whether such interactions vary between older adults in Hong Kong (HK) and older adults in the United Kingdom (UK). Altogether, 349 participants (178 HK, 171 UK; 157 males, 192 female) aged 50 years or above (mean age = 61.84 ± 8.46 years old) completed an online assessment of: (1) leisure-time physical activity (Godin–Shephard Leisure-Time Physical Activity Questionnaire); (2) personality traits (Big Five Inventory); and (3) SWB (Satisfaction with Life Scale, Positive and Negative Affect Schedule). Results showed that agreeableness, conscientiousness, extraversion, neuroticism, openness to experience, and physical activity were all significantly related to SWB in the expected direction. The relationship between physical activity and SWB was moderated by extraversion and by openness to experience: higher levels of these two traits significantly enhanced the relationship. None of the interactions varied between the HK and UK samples. The expected negative relationship between neuroticism and SWB, however, was significantly stronger in the UK sample than in the HK sample. The findings of the present study indicate that personality needs to be considered when promoting and providing physical activity for older adults, although more research is needed to further explore how this can work effectively.

## 1. Introduction

Subjective wellbeing (SWB) is defined as “*a person’s cognitive and affective evaluations of his or her life*” [1] (p. 63). The cognitive side usually refers to one’s judgments regarding life satisfaction and fulfilment, whereas the affective side typically refers to one’s emotions, moods, and feelings [1]. Improving older adults’ SWB can potentially contribute to a longer and healthier life [2,3,4,5,6]. The present study examined the primary research question of whether two well-established predictors of SWB, namely, leisure-time physical activity and personality traits, interact with each other to influence the SWB of older adults. A secondary research question examined whether such interactions varied between participants in Hong Kong (HK) and those in the United Kingdom (UK). The age of 50 is often used by health organisations as the age cut-off for physical activity programmes aimed at older adults [7], so we follow this criterion here.

### 1.1. Physical Activity and Subjective Wellbeing

Based on a large sample of 19,978 individuals (aged 18 to 64 years) from 28 European countries, Gerovasili et al.’s [8] study demonstrated that weekly physical activity levels decreased with increasing age. Nevertheless, there is evidence that physical activity in older adults can be increased effectively through a range of psychological approaches, such as goal setting [9] and motivational interviewing [10].

The positive relationship between physical activity and SWB has been well documented in cross-sectional studies targeting older adults [11,12,13,14], some of which were carried out in HK and the UK. Based on 102 adults in HK (aged 65 years or above), Poon and Fung [15] found that physical activity was positively related to satisfaction with relationships. Withall et al.’s [16] findings based on 228 adults in the UK (aged 70–96 years) showed that physical activity was positively associated with social, physical, mental, and developmental wellbeing, all assessed by the Ageing Well Profile [17], a multi-scale measure of SWB. Moreover, the impact of physical activity on SWB has been tested in longitudinal studies. For example, Ku et al.’s [18] study on 1268 participants (aged 70 years or above) reported that leisure-time physical activity measured in 1999 (T1) positively predicted subsequent SWB measured later in 2003 (T2) and in 2007 (T3). Potocnik and Sonnentag’s [19] study on 2813 retirees (aged 50 years or above) demonstrated that sports participation significantly improved quality of life over a two-year period. The findings from longitudinal research suggest that the relationship between physical activity and SWB may be causal.

There are a number of mechanisms that may explain the association between increased physical activity with higher affective and cognitive self-evaluations. In terms of the affective side, neuropsychological findings suggest that physical activity is linked with neurotransmitter release, including norepinephrine, serotonin, endorphin, and dopamine, all of which are hormones and chemicals consistently linked with emotions [20]. As for the cognitive side, having a physically active lifestyle can allow older adults to fulfil the need for time structure, meet other individuals with similar passions, and maintain close relationships with families and friends, all being important for life satisfaction [15,21]. The literature thus indicates that physical activity is positively linked with SWB.

### 1.2. Personality and Subjective Wellbeing

The Five Factor Model [22] is considered one of the most popular approaches to study personality [23]. This model emphasises five stable personality traits (also known as the Big Five) that are commonly found across people in different age groups and in different cultures. They are: extraversion, agreeableness, conscientiousness, neuroticism, and openness to experience [22,24]. Personality is widely accepted as one of the strongest and most consistent predictors of SWB [25]. According to a meta-analytic review on the relationship between the Big Five and SWB, neuroticism tends to be associated with lower SWB, whereas agreeableness, conscientiousness, extraversion, and openness to experience all tend to be associated with higher SWB [26].

The relationship between personality traits and SWB is evident in empirical studies targeting older adults. For example, based on 788 Lithuanians aged above 50 years old, Jurkuvėnas et al. [27] found that neuroticism was negatively related to SWB, whereas extraversion, agreeableness, openness to experience, and conscientiousness were all positively related to SWB. A number of recent studies have also reported similar findings to support the link between the Big Five and SWB among elderly people [28,29,30,31]. The literature thus indicates that older adults with low levels of neuroticism and high levels of agreeableness, conscientiousness, extraversion, and openness to experience are likely to report higher SWB.

### 1.3. Personality As a Potential Moderator of the Relationship Between Physical Activity and Subjective Wellbeing

Despite the accumulated evidence on: (1) the association between physical activity and SWB; and (2) the association between personality and SWB reviewed above, there has been little research on the interaction of physical activity and personality on older adults’ SWB. Studying this interaction can potentially allow practitioners and policy makers with expertise in various areas of applied psychology to have a new perspective on how to support different kinds of older adults and help them age well. 

The present study proposes that such interaction may exist because leisure-time physical activity is often considered a coping strategy for older adults—it can bring out a range of therapeutic benefits such as enhancing somatic awareness, improving sleep quality, and increasing fitness to fight disease and stress [32]. It has also been suggested that coping and personality can jointly influence SWB because personality can play a large role in how well a given coping strategy works for an individual [33]. In other words, whether physical activity can work well as a form of coping may be dependent on personality. For example, agreeable individuals tend to be more skilful at obtaining social support [34]; neurotic individuals are more likely to be exposed to interpersonal stress [33]; and conscientiousness, extraversion, and openness all relate to perceiving events as challenges rather than threats [34,35]. Taking physical activity as a coping strategy, people with high agreeableness may be more likely to receive the support they need from the people they exercise with; and people with low neuroticism may be less likely to experience conflicts with the other parties involved in their physical activity participation. When dealing with a possible setback in physical activity, individuals who are more conscientious, extraverted, or open to experiences are likely to take an active and enthusiastic approach to deal with it. All of the above examples converge to support the assumption that personality may influence the effectiveness of physical activity as a coping strategy.

Although not everyone may use leisure-time physical activity as a coping strategy, individuals with different personality traits may still experience physical activity differently [36]. For example, based on a sample of mixed-level sport participants, Allen and colleagues [36] found that those with higher levels of extraversion and emotional stability (low levels of neuroticism) tended to react more positively to unsuccessful outcomes in sports. Physical activity has also been linked with better social integration [37] and energy boost [38], both of which should be more valued by individuals with high levels of extraversion and openness to experiences. Furthermore, agreeableness and conscientiousness facilitate positive experience during social exchanges and in achievement-oriented situations, both potentially leading to higher SWB [39]. In relation to physical activity, agreeable older adults may be more likely to meet people with similar interests and have a larger social network [40]. As for conscientiousness, a trait that implies persistence, organisation, and self-discipline [39], participating in regular physical activity allows individuals to express this desirable trait because they have to be organised and manage their time well in order to exercise regularly. This in turn may also lead to the feeling of satisfaction. 

We thus hypothesise that individual personality traits will moderate the positive relationship between physical activity and SWB such that: (1) the relationship will be stronger among participants with higher levels of extraversion, agreeableness, conscientiousness, and openness to experience; and (2) the relationship will be stronger among participants with lower levels of neuroticism.

### 1.4. Cross-Cultural Differences between Hong Kong and the United Kingdom

Cross-cultural research on personality and wellbeing in Eastern culture and Western culture suggests that personality’s influence on wellbeing can vary between HK and the UK. Traditionally, individuals from European societies tend to show higher levels of SWB than individuals from East Asian societies [41,42,43]. According to the cognitive-experiential self-theory (CEST) [44], there are two different systems that affect the perceptions of personality: the experiential system (hedonism) and the rational system (realism). The former, which encourages the pursuit of pleasure, tends to be the norm within Western culture, and the latter, which calls for logical and realistic perceptions of the world, tends to be the norm within Eastern culture [43]. The CEST inspired a number of empirical studies aiming to explain the cross-cultural differences in wellbeing. For example, a recent study found that in comparison to Chinese people in HK, European Americans showed a higher level of positive evaluative bias (tendency to see themselves positively) when evaluating their personality (based on the Big Five) and satisfaction with life [45]. This study also showed that the European participants had higher wellbeing scores than the Chinese participants, which suggested that the differences between hedonism and realism (as highlighted by the positive evaluative bias) might affect personality’s influence on wellbeing [45].

Moreover, based on over 7000 participants from 28 societies, Fulmer and colleagues [46] demonstrated that culture amplified the effects of personality: the positive association between a given trait and SWB was stronger in cultures characterised by high levels of that specific trait. The main drawback of this research was that only three personality traits (extraversion, promotion focus, and locomotive regulatory mode) were assessed, and they were all desirable traits that had positive associations with SWB. In terms of the Big Five, high extraversion, agreeableness, conscientiousness, openness to experience, and emotional stability (or low neuroticism) are generally accepted as desirable [47]. Cross-cultural findings on personality [48,49,50] have shown that in comparison to individuals from Asian societies, individuals from European societies tend to report higher levels of extraversion, agreeableness, conscientiousness, and openness to experience, and lower levels of neuroticism. In other words, desirable traits from the Big Five are generally more prevalent in European cultures than in Asian cultures.

Based on the converging evidence reviewed, we assume that the positive relationship between a desirable trait and SWB will be stronger in the European (the UK) group than in the Asian (HK) group. Hence, we expect extraversion, agreeableness, conscientiousness, and openness to experience to be more positively related to SWB for the UK group than for the HK group. Regarding the relationship between neuroticism and SWB, because we assume that emotional stability (a desirable trait) will be more positively related to SWB for the UK group than for the HK group, we expect that neuroticism (i.e., low emotional stability), on the contrary, will have a stronger negative association with SWB for the UK group than for the HK group. Our hypotheses regarding group differences in the physical activity–SWB relationship and the physical activity–personality interaction are exploratory given the lack of relevant cross-cultural empirical research.

## 2. Materials and Methods

### 2.1. Participants

Participants were recruited through snowball sampling. For inclusion, participants had to be: (1) aged 50 years or above, (2) currently living in either HK or the UK, and (3) able to read English. The link of the online survey together with a short blurb introducing this study was posted on social networking sites and printed on flyers for advertisement. An email containing the same information was also sent to representatives of organisations working with older adults, and these representatives were invited to forward the email to their members and/or subscribers. Altogether, 349 adults (mean age = 61.84 ± 8.46 years old) took part in this study. The HK group had 178 participants (mean age = 57.39 ± 4.93 years old), whereas the UK group had 171 participants (mean age = 66.48 ± 8.88 years old). Further demographic information can be found in Table 1.

### 2.2. Measures

A cross-sectional online survey approach was used for data collection. In addition to gathering demographic information (country, age, gender, longstanding illness or disability, and highest level of qualification), the survey also included self-report measures focusing on leisure-time physical activity, personality traits, and SWB.

#### 2.2.1. Leisure-Time Physical Activity

The Godin–Shephard Leisure-Time Physical Activity Questionnaire (GSLTPAQ) [51] was used to measure leisure-time physical activity. The scale examined participants’ intensity and frequency of physical activity during a typical seven-day period. Respondents were asked to state the number of times per week that they spent more than 15 minutes of their free time doing: (1) strenuous exercise (which could increase their heart rate rapidly, such as running, football, vigorous swimming, etc.); (2) moderate exercise (which should not be exhausting, such as fast walking, easy swimming, popular and folk dancing, etc.); and (3) mild/light exercise (which should only require minimal effort, such as easy walking, yoga, golf, etc.). Each participant’s physical activity (GSLTPAQ score) was computed via the following standard formula: GSLTPAQ score = (9 × Strenuous Exercise) + (5 × Moderate Exercise) + (3 × Mild Exercise).

The GSLTPAQ has been found to have high test-retest reliability (over a two-week period) and high convergent validity from studies conducted on healthy adults [51,52].

#### 2.2.2. Personality Traits

The Big Five Inventory (BFI) [53] was used to assess participants’ personality traits. The scale contained a total of 44 items, including 8 items for extraversion, 9 items for agreeableness, 9 items for conscientiousness, 8 items for neuroticism, and 10 items for openness to experience. Responses were collected on a 5-point Likert-type scale, ranging from ‘disagree strongly’ to ‘agree strongly’.

The BFI’s validity coefficients with the NEO Five-Factor Inventory (NEO-FFI) [39] and the Traits Descriptive Adjectives (TDA) [54] range from 0.83 to 0.91 [53]. Three-month test–retest reliabilities and the alpha reliabilities of the BFI ranged from 0.80 to 0.90 and from 0.75 to 0.80, respectively [53]. The alpha reliabilities of the BFI based on all of the participants in this study were similar (see Table 2). When groups were analysed separately, the BFI tended to be more reliable for the UK group (except for neuroticism). With the only exception of conscientiousness in the HK group (0.69), the internal consistencies of all the subscales in BFI in both groups were higher than the generally accepted value of 0.70.

#### 2.2.3. Subjective Wellbeing

Two inventories were adopted to measure SWB: (1) the Satisfaction with Life Scale (SWLS) [55]; and (2) the Positive and Negative Affect Schedule (PANAS) [56].

The SWLS is a 5-item scale designed to measure the extent to which respondents are satisfied with their life in general [55]. This inventory is often used as an assessment of the cognitive component of SWB [57]. Participants’ responses were collected on a seven-point Likert-type scale, ranging from ‘strongly disagree’ to ‘strongly agree’.

In terms of validity, the SWLS is significantly correlated with the 28-item General Health Questionnaire (GHQ-28) [58]. As for reliability, Diener and colleagues (1985) [55] demonstrated that the internal consistency and the 2-month test–retest reliability of the SWLS were 0.87 and 0.82, respectively. In the present study, the internal consistency of SWLS was also 0.87 for both the HK sample and the combined sample (see Table 2). 

The PANAS is a 20-item scale developed to measure positive affect and negative affect [56]. In contrast to SWLS, which is mainly used as a measure of the cognitive component of wellbeing, the PANAS is generally accepted as a scale that taps the affective-emotional side of one’s wellbeing [59]. Each item in this inventory is a word that describes a feeling or an emotion. Positive affect and negative affect were each assessed by half of the scale (10 items). Participants were asked to indicate the extent to which they generally felt that way. Responses were collected on a 5-point Likert-type scale, ranging from ‘very slightly or not at all’ to ‘extremely’.

In terms of validity, Crawford and Henry [60] found that the PANAS’s positive affect subscale was negatively correlated with depression, anxiety, and stress, whereas its negative affect subscale was positively correlated with depression, anxiety, and stress. As for reliability, they reported that the internal consistencies of the PANAS were 0.89 for positive affect and 0.85 for negative affect [60]. In the present study, the alpha reliabilities of positive affect and negative affect were similar, ranging from 0.88 to 0.91 in the combined sample and the two sub-samples (see Table 2). Therefore, the three subscales of SWB used in this study were all highly reliable. 

Based on Diener et al.’s [1] definition of SWB, a composite measure of SWB was created via the combination of participants’ cognitive (satisfaction with life) and affective (balance between positive affect and negative affect) evaluation of their life using the following formula: SWB = Satisfaction with Life score (from SWLS) + Positive Affect score (from PANAS) − Negative Affect score (from PANAS). This formula has been previously used as a reliable measure of SWB [61,62].

### 2.3. Overview of Analyses 

Data were analysed via SPSS and Amos (version 22.0). Pearson product–moment correlations were used to assess the inter-correlations between all of the study variables. Five hierarchical multiple regressions were conducted to examine the interaction effects of leisure-time physical activity and each of the Big Five personality traits on SWB. In each regression, country, age, gender, longstanding illness or disability, and highest level of qualifications were entered as control variables in step 1. Next, physical activity and the personality trait assessed were entered as predictors in step 2. An interaction term between physical activity and the personality moderator was created and entered in step 3. As the measure of interaction terms would require more power, the combined sample of HK and UK participants was used, and the five traits were put into five separate regression models instead of one model. The latter could also help to reduce model complexity and ensure model stability. With respect to Cohen et al.’s [63] suggested guidelines, standardisation was used to reduce multicollinearity between main and interaction effects. To better understand the form of any of the significant interaction effects, separate regression lines were generated to illustrate the relationship between physical activity and SWB at high (one SD above the mean) and low (one SD below the mean) levels of the personality moderator [64,65]. Independent samples *t*-test was used to assess the mean differences of the variables between the HK group and the UK group. Multiple-group path analysis was used to test whether there would be any significant variations between the HK group and the UK group for: (1) the physical activity → SWB path, or (2) the personality → SWB paths, and 3) the physical activity × personality → SWB paths. 

## 3. Results

### 3.1. Descriptive Statistics 

The mean, the SD, and the inter-correlations of the variables assessed in the present study are shown in Table 3. The strongest correlation was found between neuroticism and SWB. As expected, neuroticism was the only personality trait that was negatively related to SWB. Extraversion, agreeableness, conscientiousness, and openness to experience were all positively related to SWB. Physical activity was also positively related to SWB. All of the above correlations were statistically significant at the *p* < 0.001 level. 

### 3.2. Moderation Analysis 

#### 3.2.1. Control Variables

As presented in step 1 from Table 4, Table 5, Table 6, Table 7 and Table 8, the five control variables accounted for 9% of variance in SWB, and three of them were found to be significantly associated with SWB. For country, older adults in the UK reported higher levels of SWB than older adults in HK. For longstanding illness or disability, participants without longstanding illness or disability reported higher levels of SWB than participants with longstanding illness or disability. For highest level of qualification, more educated older adults showed higher levels of SWB than less educated older adults. 

#### 3.2.2. Extraversion as Moderator 

As presented in Table 4, physical activity and extraversion accounted for 25% of variance in SWB, both positively associated with SWB. The physical activity × extraversion interaction term was significant, accounting for 2% of variance in SWB. The interaction plot (see Figure 1) shows that the positive relationship between physical activity and SWB was stronger among participants with high levels of extraversion. For older adults with low levels of extraversion, the relationship between physical activity and SWB was still positive but it was not statistically significant. Hence, extraversion was a moderator between physical activity and SWB in the expected direction. 

#### 3.2.3. Agreeableness as a Moderator 

As presented in Table 5, physical activity and agreeableness accounted for 25% of variance in SWB, both positively associated with SWB. The physical activity × agreeableness term was not significant, accounting for 0% of variance in SWB. Hence, the relationship between physical activity and SWB was not moderated by agreeableness. 

#### 3.2.4. Conscientiousness as Moderator

As presented in Table 6, physical activity and conscientiousness accounted for 22% of variance in SWB, both positively associated with SWB. The physical activity × conscientiousness interaction term was not significant, accounting for 1% of variance in SWB. Hence, the relationship between physical activity and SWB was not moderated by conscientiousness. 

#### 3.2.5. Neuroticism as Moderator

As presented in Table 7, physical activity and neuroticism accounted for 35% of variance in SWB, the former being positively associated with SWB and the latter being negatively associated with SWB. The physical activity × neuroticism term was not significant, accounting for 0% of variance in SWB. Hence, the relationship between physical activity and SWB was not moderated by neuroticism. 

#### 3.2.6. Openness to Experience as Moderator 

As presented in Table 8, physical activity and openness to experience accounted for 18% of variance in SWB, both positively associated with SWB. The physical activity × openness to experience interaction term was significant, accounting for 2% of variance in SWB. The interaction plot (see Figure 2) shows that the positive relationship between physical activity and SWB was stronger among participants with high levels of openness to experience. For older adults with low levels of openness to experience, the relationship between physical activity and SWB was still positive but it was not statistically significant. Hence, openness to experience was a moderator between physical activity and SWB in the expected direction. 

### 3.3. Findings for Hong Kong versus the United Kingdom 

In terms of mean differences between the two sub-samples, the means of four variables were significantly lower in the HK group as compared to the UK group, namely, physical activity (mean difference = −10.86, *p* < 0.001), conscientiousness (mean difference = −2.19, *p* < 0.001), openness to experience (mean difference = −2.02, *p* < 0.01), and SWB (mean difference = −5.60, *p* < 0.001). In the multiple-group path analysis, country was set as the grouping variable, with HK and the UK allocated to a group value of 1 and a group value of 2, respectively. Age, gender, longstanding illness or disability, and highest level of qualification remained as control variables. The physical activity → SWB path, the five personality → SWB paths, and the five physical activity × personality → SWB paths were all created and compared between the two groups. Appendix A shows how the HK group differed from the UK group on each of the 11 tested paths. None of the five physical activity × personality → SWB paths varied significantly between the samples. However, participants in HK differed from participants in the UK on the neuroticism → SWB path (*B* = −2.79, *p* < 0.05). Neuroticism had a stronger negative association with SWB for the UK group than for the HK group. The neuroticism → SWB path was the only path that varied significantly between the HK group and the UK group, as neither the other four personality → SWB paths nor the physical activity → SWB path differed significantly between the two groups.

## 4. Discussion

The purpose of the present study was to explore the interaction effects of leisure-time physical activity and Big Five personality traits on SWB of older adults, and whether they differed across Eastern (HK) and Western (UK) cultures. Physical activity and each of the Big Five personality traits were all significantly related to SWB in the expected direction. Extraversion, agreeableness, conscientiousness, and openness to experience were positively associated with SWB, and neuroticism was negatively associated with SWB, even after accounting for relevant covariates and the physical activity predictor. 

Among the Big Five, only extraversion and openness to experience showed a moderating effect on the relationship between physical activity and SWB. The positive link of physical activity with SWB was stronger among participants who were more extraverted and participants who were more open to experience. The significant interaction effects involving extraversion and openness to experience may be explained by previous findings showing that individuals with high levels of extraversion and individuals with high levels of openness both have a tendency to see events in their life as challenges instead of threats [34,35]. For example, after picking up an injury from exercising, those who are more extraverted may speak to different people with similar experiences to find out how they can recover as soon as possible, and they may see this recovery process as a meaningful challenge. Similarly, those who are more open to experience may explore how they can change their tactics to play a sport in order to minimise the impact of the injury on their performance (e.g., making less sprints within a football game) and/or adjust their exercise routine to prevent future injuries (e.g., spending more time on stretching and bone-strengthening activities), and such adjustments may be viewed as a positive challenge for both participants of competitive sports and participants of non-competitive exercises. Hence, both extraversion and openness to experience can play a part in how physical activity affects the way people think and feel about their lives. Another possible explanation may be relevant to physical activity’s contribution to social harmony, social growth, and energy boost [37,38]. These benefits may be more likely to be maximised by older adults who are more extraverted or more open to experience, as they tend to enjoy spending their time and energy on interacting with others and discovering new possibilities.

Agreeableness, conscientiousness, and neuroticism did not moderate the relationship between physical activity and SWB. This may be explained by research on personality and health showing how these three personality traits are most highly associated to health-related behaviours: high levels of agreeableness and conscientiousness and low levels of neuroticism are consistently linked with more health-promoting behaviours (e.g., healthy diet) and less health-harming behaviours (e.g., alcohol abuse) [66]. In other words, individuals who are agreeable, conscientious, and emotionally stable tend to be capable of maintaining their wellbeing through various ways. Therefore, physical activity itself may not have any heightened benefit for them. Moreover, it is also possible that the benefits of being agreeable, conscientious, and emotionally stable are more relevant for participants of highly competitive team sports than for participants of leisure-time physical activities that are not competitive. For example, these three traits may help individuals to co-operate with their teammates and coach, set themselves demanding goals, and remain calm in tense situations during a sport tournament, but they are less likely to make a notable difference in how older adults experience jogging, easy swimming, or walking. Hence, agreeableness, conscientiousness, and neuroticism may not necessarily have a significant influence on how older adults experience the relationship between physical activity and SWB.

Regarding the secondary analysis focusing on cross-cultural comparisons, we found that the relationship between neuroticism and SWB differed significantly between the HK sample and the UK sample—the negative relationship between neuroticism and SWB was stronger in the UK group. This may be due to neuroticism being a trait that does not fit well into the experiential system (hedonism), the more popular system within European societies which encourages people to focus on the positives [43]. Physical activity, extraversion, agreeableness, conscientiousness, and openness to experience’s relationship with SWB did not vary between the two groups. Similarly, the interaction effects of physical activity with each of the Big Five personality traits on SWB did not differ between the two groups either. This supported the decision of combining the HK group and the UK group to create a larger sample when testing the primary research question. Overall, the lack of cross-cultural differences on the tested paths may be explained by HK’s colonial ruling by the British government for over 150 years (between 1842 to 1997). HK and the UK may therefore share more cultural similarities than other Eastern and Western societies.

### 4.1. Implications

Based on the findings in this study, three main implications are suggested. Firstly, exercise psychologists and health psychologists can consider using personality screening to increase the effectiveness of some of the existing exercise-based interventions and/or physical activity promotion programmes for older adults. Secondly, according to longitudinal ageing research, personality can still be developed in old age [67]. Therefore, practitioners can encourage older participants of exercise classes to also take part in activities for personality development. This can be beneficial for them not just because this study suggests that those who are more extraverted and those who are more open to experience are likely to experience a stronger positive association between physical activity and SWB, but also because it shows that the more desirable traits themselves are all significantly associated with better SWB. Thirdly, given that employees’ SWB is often linked with their job performance [68,69], our findings may be applied in the field of occupational psychology to monitor and promote SWB among employees. Older adults with higher SWB are likely to live a longer and healthier life [2,3,4,5,6], and therefore would contribute longer to the workforce.

### 4.2. Strengths and Limitations

In terms of strengths, the three main measures of this study, including leisure-time physical activity, personality traits, and SWB, were all assessed by established inventories with good validity and reliability. More importantly, the moderation effects of extraversion and openness to experience on the relationship between physical activity and SWB found in this study offer a new perspective on how personality can be considered when promoting and providing physical activity for older adults, which may be useful for both researchers and practitioners in relevant fields of applied psychology. 

However, a few limitations need to be considered. First of all, this is a cross-sectional study, so it is not possible to address causality. Aside from that, it uses an online survey with snowball sampling for data collection, and hence the sample may be biased towards Internet users and its representativeness may not be ideal. This limitation may be reflected in that nearly 70% of the sample had completed at least one degree at university (see Table 1), whereas the percentage of older adults with degree-level or above as their highest level of qualification was lower than 30% in both HK and the UK according to census analysis reports [70,71]. In addition, all responses are self-reported, potentially leading to social desirability bias. 

### 4.3. Future Research

To address the above limitations, future research can consider using longitudinal randomised control trials to establish causality (i.e., to test whether SWB is predicted by the interaction of physical activity and personality). 

Moreover, the quantitative data themselves do not offer sufficient explanation on how and why the relationship between physical activity and SWB may be moderated by extraversion and by openness to experience. Hence, the findings of this study may be complemented by qualitative research focusing on how and why people with a certain trait may experience leisure-time physical activity differently. This type of research can also allow the researcher to explore relevant issues like: (1) whether there are personality traits other than the Big Five that may also play a part in how regular exercisers enjoy and benefit from their participation in physical activity; and (2) whether there are any cultural differences between Eastern and Western societies in terms of how physical activity is perceived by older adults.

Additionally, it is worth investigating what can be done to support older adults with low levels of extraversion and/or low levels of openness to experience, especially considering that the relationship of physical activity and SWB was not significant for them. Hence, further research is needed to explore whether and what other factors may interact with personality to influence older adults’ wellbeing.

## 5. Conclusions

To conclude, this study showed that the positive relationship between physical activity and SWB was moderated by extraversion and by openness to experience. This relationship was stronger among more extraverted older adults and among those who were more open to experience. The interaction effects between physical activity and each of the Big Five personality traits on SWB did not vary between the HK group and the UK group, although one personality trait, neuroticism, was more negatively related to SWB for the UK group than for the HK group. The findings may possibly be applied in exercise psychology, health psychology, and occupational psychology. Qualitative studies on how older adults experience physical activity in relation to their personality and wellbeing are recommended.

## Figures and Tables

**Figure 1 behavsci-08-00071-f001:**
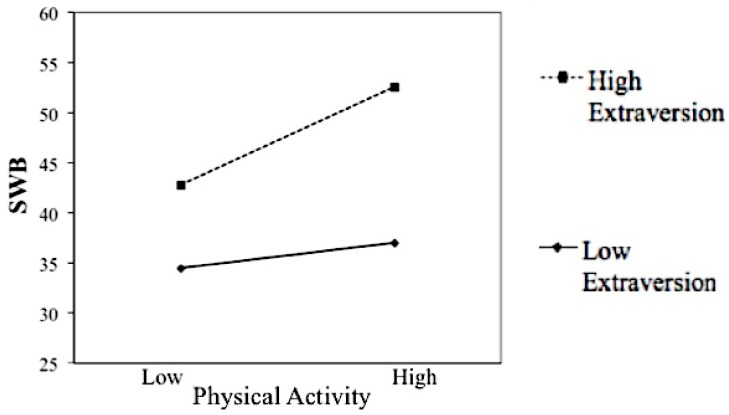
Extraversion’s moderation of the relationship between physical activity and SWB.

**Figure 2 behavsci-08-00071-f002:**
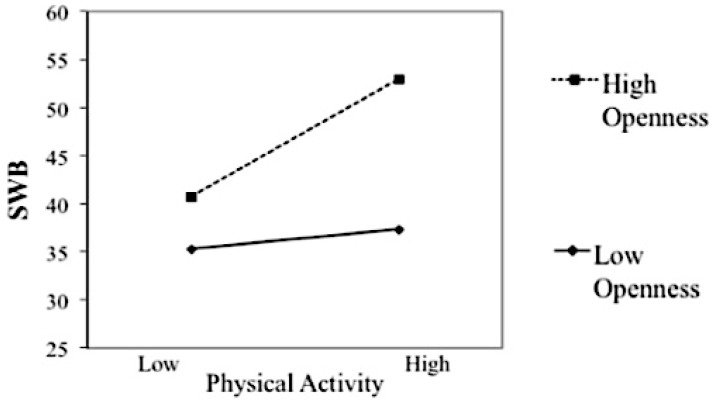
Openness to experience’s moderation of the relationship between physical activity and SWB.

**Table 1 behavsci-08-00071-t001:** Demographic distribution.

	Frequency		
	HK (*n* = 178)	UK *(n* = 171)	Total (*N* = 349)
**Gender**			
Male	66 (37.1%)	91 (53.2%)	157 (45.0%)
Female	112 (62.9%)	80 (46.8%)	192 (55.0%)
**Highest Level of Qualification**			
None	0 (0%)	12 (7.0%)	12 (3.4%)
GCSE/HKCEE/O-level	12 (6.7%)	40 (23.4%)	52 (14.9%)
A-level	13 (7.3%)	28 (16.4%)	41 (11.7%)
Undergraduate degree	61 (34.3%)	46 (26.9%)	107 (30.7%)
Postgraduate degree	86 (48.3%)	38 (22.2%)	124 (35.5%)
Doctoral degree	6 (3.4%)	7 (4.1%)	13 (3.7%)
**Longstanding Illness or Disability**			
Yes	37 (20.8%)	82 (48.0%)	119 (34.1%)
No	141 (79.2%)	89 (52.0%)	230 (65.9%)

Note. GCSE = General Certificate of Secondary Education. HKCEE = Hong Kong Certificate of Education Examination. O-level = General Certificate of Education: Ordinary Level. A-level = General Certificate of Education: Advanced Level, or Hong Kong Advanced Level Examination. Longstanding illness or disability = any physical or mental impairment, illness, or disability that has troubled the participant over a period of at least 12 months. HK: Hong Kong; UK: United Kingdom.

**Table 2 behavsci-08-00071-t002:** Cronbach’s alpha scores of personality and subjective wellbeing (SWB) measures.

	Cronbach’s Alpha		
	HK (*n* = 178)	UK (*n* = 171)	Total (*N* = 349)
Personality			
Extraversion	0.72	0.86	0.80
Agreeableness	0.78	0.79	0.77
Conscientiousness	0.69	0.81	0.77
Neuroticism	0.83	0.82	0.82
Openness to Experience	0.79	0.83	0.81
SWB			
Satisfaction with Life	0.87	0.88	0.87
Positive Affect	0.90	0.91	0.91
Negative Affect	0.90	0.88	0.89

**Table 3 behavsci-08-00071-t003:** Mean, SD, and Pearson inter-correlations for study variables.

Variable	Mean	SD	1	2	3	4	5	6	7	8	9	10	11
1. Country	1.49	0.50	−										
2. Age	61.84	8.46	0.54 ***										
3. Gender	1.55	0.50	−0.16 **	−0.21 ***									
4. Longstanding Illness or Disability	1.34	0.48	0.29 ***	0.23 ***	−0.07								
5. Highest Level of Qualification	3.91	1.23	−0.36 ***	−0.38 ***	0.07	−0.11 *							
6. Physical Activity	34.16	26.42	0.21 ***	−0.01	−0.17 **	−0.11 *	0.08						
7. Extraversion	26.53	5.81	0.09	−0.01	0.02	−0.09	0.10	0.20 ***					
8. Agreeableness	35.47	5.34	−0.07	−0.09	0.16 **	−0.13 *	0.07	0.08	0.28 ***				
9. Conscientious-ness	35.23	5.28	0.21	0.07	0.01	−0.05	−0.06	0.21 ***	0.30 ***	0.34 ***			
10. Neuroticism	19.99	5.96	0.01	−0.06	0.05	0.14 *	−0.13 *	-0.11 *	−0.27 ***	−0.43 ***	−0.33 ***		
11. Openness to Experience	35.91	6.27	0.16 **	0.03	−0.09	0.06	0.23 **	0.17 **	0.42 ***	0.19 ***	0.31 ***	−0.19 ***	
12. SWB	42.46	14.55	0.19 ***	0.08	−0.05	−0.09	0.10	0.36 ***	0.49 ***	0.43 ***	0.46 ***	−0.57 ***	0.39 ***

Note. *N* = 349. For country, 1 = HK, 2 = UK. For gender 1 = male, 2 = female. For longstanding illness or disability, 1 = no, 2 = yes. For highest level of qualification 1 = none, 2 = GCSE/HKCEE/O-level, 3 = A-level, 4 = undergraduate degree, 5 = postgraduate degree, 6 = doctoral degree. * *p* < 0.05, ** *p* < 0.01, *** *p* < 0.001.

**Table 4 behavsci-08-00071-t004:** Interaction analysis for physical activity and extraversion.

		*B*	*SE*	*β*	*R* ^2^	Δ*R*^2^
**Step 1**	Country	4.18	0.93	0.29 ***	0.09	0.09
	Age	0.53	0.92	0.04		
	Gender	−0.26	0.77	−0.02		
	Longstanding Illness or Disability	−2.39	0.78	−0.16 **		
	Highest Level of Qualification	2.84	0.82	0.20 **		
**Step 2**	Country	1.76	0.84	0.12 *	0.34	0.25
	Age	1.17	0.80	0.08		
	Gender	0.11	0.67	0.01		
	Longstanding Illness or Disability	−1.07	0.69	−0.07		
	Highest Level of Qualification	1.46	0.72	0.10 *		
	Physical Activity	3.50	0.70	0.24 ***		
	Extraversion	6.30	0.69	0.42 ***		
**Step 3**	Country	1.41	0.84	0.10	0.35	0.02
	Age	1.68	0.81	0.12 *		
	Gender	0.40	0.67	0.03		
	Longstanding Illness or Disability	−0.90	0.68	−0.06		
	Highest Level of Qualification	1.48	0.71	0.10 *		
	Physical Activity	3.08	0.71	0.21 ***		
	Extraversion	5.96	0.69	0.39 ***		
	Interaction (Physical Activity × Extraversion)	1.79	0.62	0.14 **		

Note. *N* = 349. *B* = unstandardised beta. *SE* = standard error. *β* = standardised beta. Δ*R*^2^ = change in *R*^2^. * *p* < 0.05, ** *p* < 0.01, *** *p* < 0.001.

**Table 5 behavsci-08-00071-t005:** Interaction analysis for physical activity and agreeableness.

		*B*	*SE*	*β*	*R* ^2^	Δ*R*^2^
**Step 1**	Country	4.18	0.93	0.29 ***	0.09	0.09
	Age	0.53	0.92	0.04		
	Gender	−0.26	0.77	−0.02		
	Longstanding Illness or Disability	−2.39	0.78	−0.16 **		
	Highest Level of Qualification	2.84	0.82	0.20 **		
**Step 2**	Country	2.68	0.83	0.18 **	0.34	0.25
	Age	1.29	0.80	0.09		
	Gender	−0.48	0.67	−0.03		
	Longstanding Illness or Disability	−1.02	0.69	−0.07		
	Highest Level of Qualification	2.05	0.71	0.14 **		
	Physical Activity	3.90	0.69	0.27 ***		
	Agreeableness	6.72	0.73	0.41 ***		
**Step 3**	Country	2.65	0.83	0.18 **	0.34	0.00
	Age	1.33	0.80	0.09		
	Gender	−0.48	0.68	−0.03		
	Longstanding Illness or Disability	−0.98	0.69	−0.07		
	Highest Level of Qualification	2.05	0.71	0.14 **		
	Physical Activity	3.81	0.72	0.26 ***		
	Agreeableness	6.71	0.74	0.41 ***		
	Interaction (Physical Activity × Agreeableness)	0.34	0.77	0.02		

Note. *N* = 349. *B* = unstandardised beta. *SE* = standard error. *β* = standardised beta. Δ*R*^2^ = change in *R*^2^. ** *p* < 0.01, *** *p* < 0.001.

**Table 6 behavsci-08-00071-t006:** Interaction analysis for physical activity and conscientiousness.

		*B*	*SE*	*β*	*R* ^2^	Δ*R*^2^
**Step 1**	Country	4.18	0.93	0.29 ***	0.09	0.09
	Age	0.53	0.92	0.04		
	Gender	−0.26	0.77	−0.02		
	Longstanding Illness or Disability	−2.39	0.78	−0.16 **		
	Highest Level of Qualification	2.84	0.82	0.20 **		
**Step 2**	Country	1.37	0.86	0.09	0.31	0.22
	Age	1.28	0.81	0.09		
	Gender	0.14	0.68	0.01		
	Longstanding Illness or Disability	−1.11	0.70	−0.08		
	Highest Level of Qualification	2.29	0.73	0.16 **		
	Physical Activity	3.57	0.71	0.25 ***		
	Conscientiousness	6.32	0.77	0.39 ***		
**Step 3**	Country	1.21	0.86	0.08	0.32	0.01
	Age	1.54	0.83	0.11		
	Gender	0.12	0.68	0.01		
	Longstanding Illness or Disability	−1.00	0.70	−0.07		
	Highest Level of Qualification	2.29	0.72	0.16 **		
	Physical Activity	3.20	0.75	0.22 ***		
	Conscientiousness	6.31	0.77	0.39 ***		
	Interaction (Physical Activity × Conscientiousness)	1.23	0.81	0.08		

Note. *N* = 349. *B* = unstandardised beta. *SE* = standard error. *β* = standardised beta. Δ*R*^2^ = change in *R*^2^. ** *p* < 0.01, *** *p* < 0.001.

**Table 7 behavsci-08-00071-t007:** Interaction analysis for physical activity and neuroticism.

		*B*	*SE*	*β*	*R* ^2^	Δ*R*^2^
**Step 1**	Country	4.18	0.93	0.29 ***	0.09	0.09
	Age	0.53	0.92	0.04		
	Gender	−0.26	0.77	−0.02		
	Longstanding Illness or Disability	−2.39	0.78	−0.16 **		
	Highest Level of Qualification	2.84	0.82	0.20 **		
**Step 2**	Country	2.72	0.76	0.19 ***	0.45	0.35
	Age	−0.09	0.74	−0.01		
	Gender	0.65	0.61	0.05		
	Longstanding Illness or Disability	−0.49	0.63	−0.03		
	Highest Level of Qualification	0.91	0.66	0.06		
	Physical Activity	3.88	0.63	0.27 ***		
	Neuroticism	−8.45	0.65	−0.54 ***		
**Step 3**	Country	2.61	0.78	0.18 **	0.45	0.00
	Age	−0.01	0.75	0.00		
	Gender	0.66	0.61	0.05		
	Longstanding Illness or Disability	−0.46	0.63	−0.03		
	Highest Level of Qualification	0.89	0.66	0.06		
	Physical Activity	3.81	0.64	0.26 ***		
	Neuroticism	−8.50	0.66	−0.54 ***		
	Interaction (Physical Activity × Neuroticism)	−0.49	0.72	−0.03		

Note. *N* = 349. *B* = unstandardised beta. *SE* = standard error. *β* = standardised beta. Δ*R*^2^ = change in *R*^2^. ** *p* < 0.01, *** *p* < 0.001.

**Table 8 behavsci-08-00071-t008:** Interaction analysis for physical activity and openness to experience.

		*B*	*SE*	*β*	*R* ^2^	Δ*R*^2^
**Step 1**	Country	4.18	0.93	0.29 ***	0.09	0.09
	Age	0.53	0.92	0.04		
	Gender	−0.26	0.77	−0.02		
	Longstanding Illness or Disability	−2.39	0.78	−0.16		
	Highest Level of Qualification	2.84	0.82	0.20 **		
**Step 2**	Country	1.45	0.89	0.10	0.27	0.18
	Age	1.16	0.84	0.08		
	Gender	0.74	0.70	0.05		
	Longstanding Illness or Disability	−1.76	0.72	−0.12 *		
	Highest Level of Qualification	0.70	0.78	0.05		
	Physical Activity	4.07	0.73	0.28 ***		
	Openness to Experience	5.32	0.81	0.33 ***		
**Step 3**	Country	1.42	0.88	0.10	0.29	0.02
	Age	1.24	0.83	0.09		
	Gender	0.96	0.69	0.07		
	Longstanding Illness or Disability	−1.78	0.71	−0.12 *		
	Highest Level of Qualification	0.63	0.77	0.04		
	Physical Activity	3.57	0.73	0.25 ***		
	Openness to Experience	5.25	0.80	0.32 ***		
	Interaction (Physical Activity × Openness to Experience)	2.57	0.76	0.16 **		

Note. *N* = 349. *B* = unstandardised beta. *SE* = standard error. *β* = standardised beta. Δ*R*^2^ = change in *R*^2^. * *p* < 0.05, ** *p* < 0.01, *** *p* < 0.001.

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
