# Peer review of "Interaction of Physical Activity and Personality in the Subjective Wellbeing of Older Adults in Hong Kong and the United Kingdom"

_behavsci, 2018, doi:10.3390/bs8080071_

Round 1

Reviewer 1 Report

The paper is about three well-known measurement constructs: the Big Five, SWB, and physical activity and their interplay. Instead of just relating the Big Five and physical activity with SWB separately, the interaction between the Big Five and physical ability and its influence on SWB are analyzed. In addition, it was applied to two different countries (HongKong and UK) and to a special group of people aged >=50.

The measurement constructs, Big Five Inventory, Godin-Shephard Leisure-Time Physical Activity Questionnaire, and the SWB items are very well chosen and properly described in the “2.2 Measures” section.

The literature review is well worked out, to be precise, the “1.1 Physical activity and subjective wellbeing” section lists country related results for UK and HongKong, also the moderating effect of the Big Five on the influence of physical activity on SWB is discussed and motivated by the work of several other authors. The literature ends up with cultural differences between East and West studies from the wellbeing perspective (hedonism vs. realism) etc. Motivation for the study at hand is given and well introduced!

One or two sentences on the online snowball sampling would be welcome. How were the first respondents contacted? Where were the addresses taken from?

The use of several control variables is also well done.

The stepwise development of the models is well thought out: 1) control variables, 2) just predictors, 3) predictors + interaction.

There is a guideline mentioned to prevent multicollinearity, namely standardization. I wonder how standardization can eliminate the multicollinearity of the predictors. However, it can reduce multicollinearity between main and interaction effects. This should be mentioned. Furthermore, it would be nice to see variance inflation factor (VIF) and/or tolerance values (as SPSS was used) for the reader to be able to evaluate multicollinearity between the predictors on his/her own.

Or at least one could discuss them next to the correlation Table 3.

Multiple-group path analysis to take into consideration the country of origin is well chosen.

Bad resolution of Table 3. I cannot read most of the values. -> Please do not take a screenshot but insert the original table.

Results and interpretation of results are done very well. Also the multi-group comparison between UK and HongKong at the end of the results section (with the table in the appendix) is well done.

However, it should be argued why all Big Five traits were put into separate regression models and not all of them were also for comparison purposes put into one model like in the very last one of the results chapter comparing the two groups. The five single trait models [already in the paper] vs. the combined model [physical activity -> SWB (main effect), all 5 personality traits -> SWB (main effects), all 5 interaction effects between the physical activity and the 5 personality traits] would also be interesting to see.

The discussion chapter is very well done. I also liked the interpretation from the hedonism vs. realism point of view.

Author Response

Thanks for the comments. I made the following changes after considering your suggestions.

Comment 1: One or two sentences on the online snowball sampling would be welcome. How were the first respondents contacted? Where were the addresses taken from?

Response: In section 2.1, I added more information about the online snowball sampling: “Participants were recruited through snowball sampling. The inclusion criteria were: 1) aged 50 years or above, 2) currently living in either HK or the UK, and 3) able to read English. The link of the online survey together with a short blurb introducing this study was posted on social networking sites and printed on flyers for advertisement. An email containing the same information was also sent to representatives of organisations working with older adults, and these representatives were invited to forward the email to their members and/or subscribers.

Comment 2: Standardization can reduce multicollinearity between main and interaction effects. This should be mentioned.

Response: In section 2.3, I added that “standardisation was used to reduce multicollinearity between main and interaction effects”.

Comment 3: Bad resolution of Table 3. I cannot read most of the values. -> Please do not take a screenshot but insert the original table.

Response: The screenshot for table 3 was removed, and the original table was inserted.

Comment 4: It should be argued why all Big Five traits were put into separate regression models and not all of them were also for comparison purposes put into one model like in the very last one of the results chapter comparing the two groups.

Response: In section 2.3, I added the following text: “As the measure of interaction terms would require more power, the combined sample of HK and UK participants was used, and the five traits were put into five separate regression models instead of one model. The latter could also help to reduce model complexity and ensure model stability”. 

Reviewer 2 Report

Congratulations you have written and conducted an important research project. Comparison studies add more information concerning different cultures and countries. The entire issue of expanding the information concerning SWB of seniors is helpful for all populations. I suggest in your conclusions, and maybe elsewhere, to discuss something about the implications for exercise classes. So much of your information could be used within the context of an exercise class where you encourage the participants to also take part in the personality development. 

Author Response

Thanks for the comments! I'll add the point about the implications for exercise classes in my discussion.

Reviewer 3 Report

General comments

1.       Much of the literature seems old. Noted in particular were old references about physical activity (PA) research (eg references 7, 8, 9, 13, 14, 17, 24, 25 etc). There has been quite a bit of research in the last 5 years about PA in older adults, and the authors are encouraged to reference the latest research.

2.       The terms ‘physical activity’ and ‘exercise’ seemed to be used interchangeably. Exercise is a subset of physical activity (PA). Also, it is important to be clear about what type of PA is being studied. For example, PA could be leisure-time PA (LTPA); health-enhancing LTPA (meets guidelines for PA), or total PA (including work-related PA and PA for travel). Which type was examined in this study? The measure used was originally developed to measure exercise lasting at least 15 minutes, and it appears that exercise was measured based on the questions.

3.       The literature section would be strengthened with more support from the literature for statements being made. For example, in the paragraph from line 50, it is noted that the relationship between PA and SWB is well-documented. However, only 2 (older) studies are included with two additional ones provided as example. Later, the authors conclude that studies of longitudinal associations indicate causal relationship but only 2 studies are cited, and evidence of causality requires more than findings from longitudinal studies. In short, provide more citations to show a better review of the literature. The studies don’t all have to be discussed. Also, be clear where results are consistent across studies and where results are mixed (and be careful in stating causality).

4.       The Introduction makes the argument that the interaction between PA and SWB has not been adequately studied in older adults and mechanisms that would explain the relationship are presented. However, missing is why the interaction is important to know, and the introduction would be strengthened with such an argument.

5.       Line 169 says that participants were recruited through online snowball sampling. Additional details about the sampling are required. Was there a sampling frame? Was the same sampling from used in both countries. Why or why not?  How did the snowballing happen? Was there any additional inclusion/exclusion criteria? For example, were people who are unable to do physical activity excluded? Did people have to be community-dwelling or were people in residential aged care facilities included? The use of a snowball technique should be discussed as a limitation in Section 4.2.

6.       Line 178: What was the question that was asked to determine ‘longstanding illness or disability’? If such illnesses or disabilities were ones that impacted upon ability to do PA, what would be the rationale for including these respondents in the analyses? 

7.       Line170: What was the age of the two samples separately? Are there data available on race/ethnicity?

8.       The authors are encouraged to look at the possibility of reverse-causality (eg well-being becomes the independent variable and PA the dependent variable).

9.       Section 2.3. What is the rationale for putting the different personality subscales into different models?

10.   The implications and future research sections suggest that the results can be used for influencing practice and seem to go beyond what this cross-sectional study of a small, volunteer sample can provide. Therefore, the authors are encouraged to be more cautious in discussing the next steps for this line of research and its relevance to practice.

Minor comments

11.   Wording saying “more negatively related” in various place could be improved (e.g., “ stronger negative association”?)

12.   Lines 199-200: For readers not familiar with the inventories discussed, it would be useful to describe them here.

13.   Line 217: Table 2 does not show the estimates to be ‘exactly the same’.

14.   Line 231: Should this be changed from 0.87 to 0.88?

15.   Line 384: Consider using an example more relevant to the population.

16.   Line 400: Clarify why PA would not be a health-promoting behavior that is habituated.

17.   Line 421: This conclusion does not appear to be supported.

Author Response

Thank you for the detailed comments. I made the following changes after considering your suggestions.

Comment 1: Much of the literature seems old. Noted in particular were old references about physical activity (PA) research.

Response: Older references (no. 7, 8, 9, 13, 17 and 24 from the original version) were removed and replaced by more updated references from the past 4 years (all of the newly added references were highlighted in the references list in the end).

Comment 2: It is important to be clear about what type of PA is being studied.

Response: I emphasised that I'm focusing on leisure-time physical activity rather than just any physical activities.

Comment 3: In the paragraph from line 50, it is noted that the relationship between PA and SWB is well-documented. However, only 2 (older) studies are included with two additional ones provided as example. 

Response: In section 1.1, the number of citations used to support how “The positive relationship between physical activity and SWB has been well documented in cross-sectional studies targeting older adults” increased from 2 in the original version to 4 in the new version.

Comment 4: The authors conclude that studies of longitudinal associations indicate causal relationship but only 2 studies are cited, and evidence of causality requires more than findings from longitudinal studies.

Response: In section 1.1, “The findings from longitudinal research suggest that the relationship between physical activity and SWB is causal.” was edited to “The findings from longitudinal research suggest that the relationship between physical activity and SWB may be causal”.

Comment 5: The Introduction makes the argument that the interaction between PA and SWB has not been adequately studied in older adults and mechanisms that would explain the relationship are presented. However, missing is why the interaction is important to know, and the introduction would be strengthened with such an argument.

Response: In section 1.3, I added the argument that “Studying this interaction can potentially allow practitioners and policy makers with expertise in various areas of applied psychology to have a new perspective on how to support different kinds of older adults and help them age well”.

Comment 6: Additional details about the sampling are required.

Response: in section 2.1, I added that “Participants were recruited through snowball sampling. The inclusion criteria were: 1) aged 50 years or above, 2) currently living in either HK or the UK, and 3) able to read English. The link of the online survey together with a short blurb introducing this study was posted on social networking sites and printed on flyers for advertisement. An email containing the same information was also sent to representatives of organisations working with older adults, and these representatives were invited to forward the email to their members and/or subscribers”.

Comment 7: The use of a snowball technique should be discussed as a limitation in Section 4.2.

Response: In section 4.2, I edited the second limitation as “it uses online survey with snowball sampling for data collection, and hence the sample may be biased towards Internet users and its representativeness may not be ideal. This limitation may be reflected in that nearly 70% of the sample completed at least one degree at university (see Table 1), whereas the percentage of older adults with degree level or above as their highest level of qualification was lower than 30% in both HK and the UK according to census analysis reports”.

Comment 8: What was the question that was asked to determine ‘longstanding illness or disability’?

Response: The question for longstanding illness or disability was "Do you have any long-standing physical or mental impairment, illness or disability ("long-standing" refers to anything that has troubled you over a period of at least 12 months or that is likely to trouble you over a period of at least 12 months)?", it was briefly explained via the note under Table 1.

Comment 9: What was the age of the two samples separately? Are there data available on race/ethnicity?

Response: In section 2.1, I added that “The HK group had 178 participants (mean age = 57.39 ± 4.93 years old), whereas the UK group had 171 participants (mean age = 66.48 ± 8.88 years old)”. Unfortunately, there're no data available on race/ethnicity.

Comment 10: What is the rationale for putting the different personality subscales into different models?

Response: In section 2.3, I added that “As the measure of interaction terms would require more power, the combined sample of HK and UK participants was used, and the five traits were put into five separate regression models instead of one model. The latter could also help to reduce model complexity and ensure model stability”.

Comment 11: The implications and future research sections suggest that the results can be used for influencing practice and seem to go beyond what this cross-sectional study of a small, volunteer sample can provide. Therefore, the authors are encouraged to be more cautious in discussing the next steps for this line of research and its relevance to practice.

Response: In section 4.3, the future research suggestion “practitioners can consider the possibility of designing tailor-made exercise groups for older adults based on their personality and testing their effectiveness. For example, those with high levels of extraversion may be more likely to enjoy taking part in the exercise group if they get to bring someone new to the group with them occasionally or even every time. Similarly, instead of playing a sport in the traditional way, it may be more satisfying for participants with high levels of openness if they can make use of their creativity and imagination to experience the sport differently” was replaced by “To address the above limitations, future research can consider using longitudinal randomised control trials to establish causality (i.e., to test whether SWB is predicted by the interaction of physical activity and personality)”.

Comment 12: Wording saying “more negatively related” in various place could be improved (e.g., “ stronger negative association”?)

Response: The two “more negatively related” (line 163 and line 364 in the original version) were replaced by “have/had a stronger negative association” (line 165 and line 381 in the new version).

Comment 13: Line 217: Table 2 does not show the estimates to be ‘exactly the same’.

Response: “exactly the same” (line 218 in the original version) was replaced by “also 0.87” (line 226 in the new version).

Comment 14: Line 231: Should this be changed from 0.87 to 0.88?

Response: “0.87” (line 232 in the original version) was replaced by "0.88" (line 240 in the new version), that was a typo, thanks for spotting it out.